



# The polar mesospheric cloud dataset of the Balloon Lidar Experiment BOLIDE

Natalie Kaifler[1], Bernd Kaifler[1], Markus Rapp[1], and David C. Fritts[2]

[1]Deutsches Zentrum für Luft- und Raumfahrt, Institut für Physik der Atmosphäre, Oberpfaffenhofen, Germany
[2]GATS, Boulder, CO, USA

**Correspondence:** Natalie Kaifler (natalie.kaifler@dlr.de)

**Abstract.** The Balloon Lidar Experiment (BOLIDE) observed polar mesospheric clouds (PMC) along the Arctic circle between Sweden and Canada during the balloon flight of PMC Turbo in July 2018. The purpose of the mission was to study small-scale dynamical processes induced by the breaking of atmospheric gravity waves by high-resolution imaging and profiling of the PMC layer. The primary parameter of the lidar soundings is the time- and range-resolved volume backscatter coefficient $\beta$. These data are available at high resolution of 20 m and 10 s (Kaifler, 2021, https://zenodo.org/record/5722385). This document describes how we calculate $\beta$ from the BOLIDE photon count data and balloon floating altitude. We compile information relevant for the scientific exploration of this dataset, including statistics, mean values and temporal evolution of parameters like PMC brightness, altitude and occurrence rate. Special emphasis is given to the stability of the gondola pointing, and the effect of resolution on the signal-to-noise ratio and thus the detection threshold of PMC. PMC layers were detected during 49.7 h in total, accounting for 36.8 % of the 5.7 days flight duration and a total of 178924 PMC profiles at 10 s resolution. Up to the present, published results from subsets of this dataset include the evolution of small-scale vortex rings, distinct Kelvin-Helmholtz instabilities and mesospheric bores. The lidar soundings reveal a wide range of responses of the PMC layer to larger-scale gravity waves and breaking gravity waves including accompanying instabilities that await scientific analysis.

## 1 Introduction

The dataset described here contains balloon lidar soundings of polar mesospheric clouds (PMC) which are also known as noctilucent clouds (NLC). Both terms name the same phenomenon, i.e. optically thin layers of ice particles at around 83 km altitude which occur at high latitudes during polar summer. The term noctilucent clouds is translated from the German term "Leuchtende Nachtwolken" coined by Otto Jesse who was among the first observers in 1885, and who was the first to recognize the scientific value of their observation regarding the exploration of what is now known as the mesosphere-lower thermosphere region (Jesse, 1885; Leslie, 1885; Backhouse, 1885). One of the prime interests in PMC research today is the structure and dynamical evolution of atmospheric gravity waves imprinted on this cloud layer, as first noted by Witt (1962). Noctilucent



clouds can be observed by naked eye, i.e. in the visible part of the electromagnetic spectrum, or by camera from ground within the twilight zone. Satellite instruments in orbit since the 1980's extend the observations to different wavelengths, all longitudes

and a wider range of latitudes, and hence the more general term polar mesospheric clouds was introduced (Thomas, 1984; DeLand et al., 2006). This term has also been used related to observations with cameras from space (Chandran et al., 2009). With the advent of lidars that often operate at 532 nm wavelength – and thus within the wavelength range that human observers can see – and that were initially ground-based, the term noctilucent clouds has been retained also for historic reasons. However, the majority of PMC data from ground-based lidars is acquired at polar latitudes in full daylight due to the higher occurrence

frequency closer to the poles. As such, the term PMC also became common in connection with ground-based lidars (e.g. Chu et al., 2003). The Balloon Lidar Experiment (BOLIDE) is the first lidar to make observations from a balloon platform in the upper stratosphere independent of the influence of the lower atmosphere, e.g. tropospheric clouds, and the mission design and operation had many aspects of a small satellite mission. For this reason, and for conformity to the mission name, we use the term PMC in connection with this dataset.

The BOLIDE instrument is a high-power Rayleigh lidar designed for operation on long duration balloons floating at ∼ 40 km altitude in the Arctic and Antarctic (Kaifler et al., 2020). The laser beam produced by a pulsed Nd:YAG laser with a mean optical power of 4.5 W and 100 Hz pulse rate is tilted 28 deg off-zenith to avoid the balloon. Backscattered light is collected by a 0.5-m-diameter mirror and detected via an avalanche photo diode. The detector is operated in photon-counting mode, i.e. every single detected photon is recorded with its time stamp relative to the emission of the last laser pulse. The native

temporal resolution of the BOLIDE data acquisition is 800 ps, but the effective resolution is limited to the duration of the laser pulses which is 5 ns or 1.5 m in range. For all practical applications, these raw photon count data are binned to lower resolution in order to achieve a sufficiently high signal-to-noise ratio. For example, Kaifler et al. (2020, their Fig. 8) show photon count data at 10 m and 1 s resolution. For scientific interpretation, photon counts are converted to volume backscatter coefficients that relate the measured atmospheric return signal to a standard atmospheric density profile. For detection of scattering originating

from PMC particles, we define a threshold of 2.5 $\sigma$ relative to the background signal. The source of this background signal is scattered sunlight and as such contributes to the lidar signal measured in every altitude range. In our opinion, when calculating volume backscatter coefficients from BOLIDE measurements, the photon count data binned to 20 m vertical and 10 s time resolution represents a good compromise between the diametral requirements high resolution and high signal-to-noise ratio. The resulting data set is of comparable quality as the much larger PMC data set produced by the Arctic Lidar Observatory for

Middle Atmosphere Research (ALOMAR) Rayleigh/Mie/Raman lidar (Kaifler et al., 2018; Schäfer et al., 2020). Moreover, Taylor et al. (2009) and Collins et al. (2009) have also analyzed PMC lidar data at resolutions below 10 min.

   The PMC Turbo mission was designed to acquire quasi-3d soundings of the PMC layer by using a combination of camera and lidar observations at high resolution, as it is only at scales below one minute and tens of meters that the fine structures imprinted by dynamical instabilities during the breaking of gravity waves and the transition to turbulence become evident (Fritts

et al., 2019). Besides the BOLIDE lidar, the scientific payload consisted of four wide-field-of-view and three narrow-field-of-view cameras (Kjellstrand et al., 2020). In this work, we describe and characterize the BOLIDE PMC dataset available from zenodo (Kaifler, 2021) and NASA's Space Physics Data Facility (see Sec. 6). The netcdf file includes the volume backscatter





coefficient $\beta$ at 20 m and 10 s resolution, as well as time series of the gondola floating altitude, the orientation of the gondola in azimuth, as well as latitude and longitude of the lidar beam at 82 km altitude.

## 2 Balloon floating altitude and gondola stabilization

PMC Turbo was carried by a long duration balloon launched at Esrange, Sweden, in July 2018. The stratospheric winds carried the instruments across the Norwegian Sea, the Greenland ice sheet, Baffin Bay and Baffin Island to the northern Kivalliq Region of Nunavut in Canada, resulting in a flight track within a latitude band of 66.33–69.46 deg north. Maps with the flight track are shown by Fritts et al. (2019). The length of the flight track was 5810 km, and the average speed was 11.7 m/s with a standard deviation of 4.3 m/s due to changing upper stratospheric winds. Of relevance for the analysis of lidar PMC data is the balloon floating altitude $z_g$ as it constitutes an offset in the calculation of the absolute PMC layer altitude based on the range data measured by the lidar. $z_g$ changes as a function of local solar time due to the thermal heating of the balloon, and occasionally ballast was dropped to gain height (Fig. 1a). $z_g$ was measured by several GPS receivers mounted on the balloon gondola carrying BOLIDE, and the data were transmitted to ground via satellite links as well as stored onboard at different temporal resolutions. These data sets have been quality-controlled, merged and interpolated to a 5 s-grid. Ascent to the upper stratosphere was completed on 7 July at 10:10 UT at 20.71 deg E and descent was initiated on 14 July 2018 at 4:46 UT at 109.63 W. The average floating altitude was 38.27 km. Minimum and maximum altitudes were 35.75 km and 39.55 km, respectively. Wavelet analyses of the floating altitude reveal the dominant diurnal component and a buoyancy oscillation of $\approx$ 5 min period and $\approx$ 40 m amplitude (Fig. 1b). A pronounced buoyancy oscillation can be seen in the top panel of Fig. 2e. The PMC altitudes in the published data set have been corrected for variations of the floating altitude.

The lidar beam was pointed $28.0 \pm 0.1$ deg off-zenith. The slanted beam resulted in the probing of a PMC layer centered at 82 km altitude at 23.25 km horizontal distance to the position of the gondola projected onto the PMC layer. Due to variations in floating altitude, this distance varied between 22.57 km and 24.59 km during the flight. The balloon's buoyancy oscillation of 40 m amplitude lead to a periodic horizontal displacement of the laser beam with an amplitude of 210 m at the altitude of the PMC layer, whereas the uncertainty in the beam pointing angle corresponds to an uncertainty in horizontal distance of approximately 80 m. Due to the off-zenith beam angle, a wide PMC layer extending 80–86 km in altitude was thus probed diagonally with the lower boundary being horizontally offset by 3 km relative to the upper boundary. For an average PMC layer width of 1 km, this value amounts to 500 m, accordingly. The azimuth angle $\alpha$ changed continuously as a rotator stabilized the pointing of the gondola in anti-sun direction, resulting in one full rotation during one local day (Fig.1c). In the fixed reference frame of the gondola, this rotation caused the laser beam to perform a circular motion at the PMC layer with approximately 24 km radius and a speed of 1.7 m/s. In the earth-centered reference frame, this circular motion is overlaid with the drift of the gondola which was on average seven times faster. A rotation of 1 deg displaced the laser beam by 406 m at 82 km altitude. The lidar was turned off during tests when the gondola was rotated towards the sun during non-PMC conditions in the evenings of 11 and 13 July. Few sudden losses of azimuth control occurred during the mission. The recovery from those control problems resulted in faster sweeps of the lidar beam over short distances. To automatically detect these events

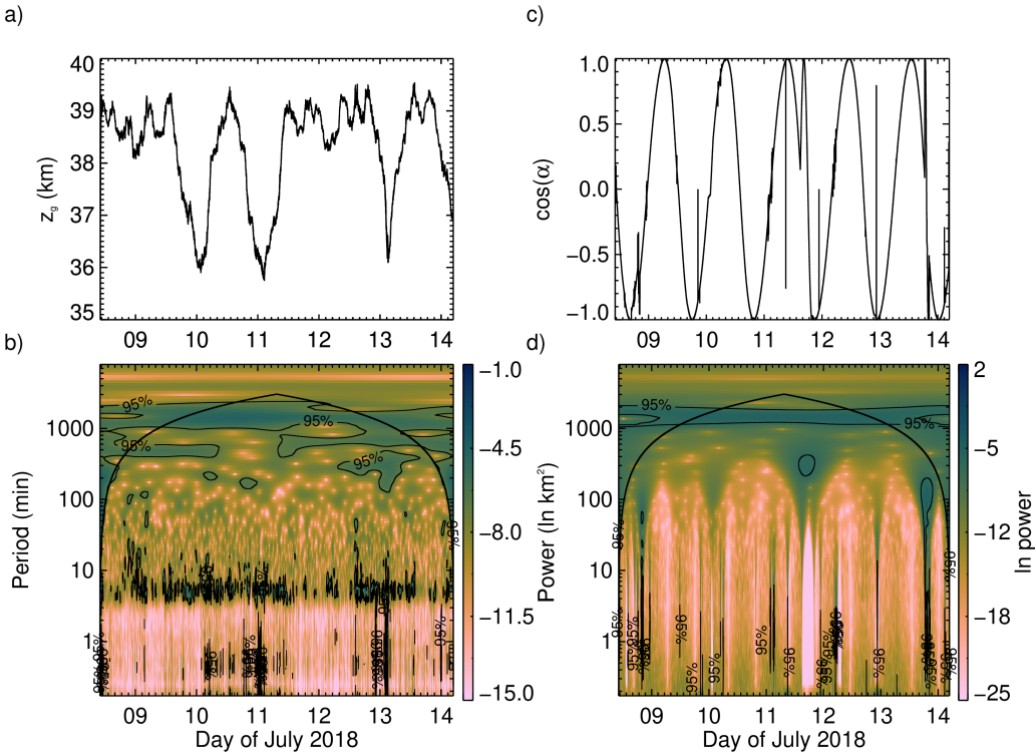

**Figure 1.** (a) Interpolated floating altitude $z_g$ and (b) its wavelet spectrum, showing a dominant diurnal motion and a buoyancy motion of 5 min period. (c) Cosine of the azimuth angle $\alpha$, and (d) its wavelet spectrum including short losses of pointing stabilization around 1 min period and tests during non-PMC conditions and prior to descent.

in post-processing, we look for periods with significant spectral power at 1 min period using wavelet analysis of the cosine of the azimuth angle (Fig. 1d). Only six of these events occured during PMC conditions: on 8 July between 13:58–14:11 UT at 16:53 UT and between 23:03–23:20 UT, on 10 July between 5:51–6:23 UT and on 11 July between 1:35–1:54 UT and 2:54–3:32 UT. Anticipating the calculation of $\beta$ which is described in the following section, we show the PMC detections at 20 m × 10 s-resolution that accompany said anomalies in the gondola pointing along with azimuth angle and floating altitude in Fig. 2. Those PMC layer observations are not invalid, but it may be necessary to account for the fast-moving laser beam when interpreting layer displacements. At the average speed of 12.1 m/s of the laser beam at the height of the PMC layer, one 10-s-lidar profile averages over a horizontal distance of 121 m. The spot size of the laser beam at the PMC layer follows from the mean divergence of the laser beam and the slant range. Using a mean divergence angle of 67 µrad (Kaifler et. al 2020), the calculated spot size is 6.3 m. The position of the lidar beam is fixed relative to the FOVs of PMC Turbo's camera systems, which have a pixel size at the position of the lidar beam of 3 m and 8 m, respectively (Kjellstrand et al., 2020). The actual resolution of features in the PMC layer depends on the local wind speed with which they are advected through the lidar's field of view. Model estimates of the wind speed derived from radar data e.g. amount to 40 m/s between 2–3 UT on 10 July

95

100



2018 (Geach et al., 2020). Applying tracking algorithms to small-scale features in series of images acquired by the on-board
cameras, the local wind speed can also be derived from measurements (Geach et al., 2020). As these analyses are inherently
complex, e.g. in case of multiple layers in sheared environments, information on relative velocities of the position of the laser
beam within the advecting PMC layers is not included as a standard product for the whole flight.

## 3  Calculation of volume backscatter coefficients

The first step in the calculation of the volume backscatter coefficient is the conversion from slant range to vertical range by
multiplying the range grid of the 20 m × 10 s-binned photon count profiles with the cosine of the zenith angle. The geometric
altitude is obtained by adding the altitude of the gondola, $z_g$, to the vertical lidar range. The second step is then the removal
of the so-called background. The background originates from solar radiation scattered into the optical path of the receiving
telescope by the residual atmosphere and the telescope's spider, and results in a range-independent offset in the lidar data.
Signal-induced noise produced by the photon detector introduces a weak range-dependence, however. For that reason, we
estimate the background $F$ above the PMC layer between 96 and 120 km altitude by fitting a linear model

$$F = A_0 + (0.017 + 3.72 \times 10^{-5} A_0) z \tag{1}$$

to each count profile. The coefficients in the linear term in Eqn. 1 were determined empirically from calibration measurements.
$A_0(t)$ ranges between 20–40 photon counts and is strongly correlated with the solar zenith angle, but is also influenced by
PMC (the flight-time series is shown below in Fig. 5a). After removal of the background, the count profiles are scaled with
$(z - z_g)^{-2}$ to correct for the range-dependent solid angle of the backscattered laser light. The resulting backscatter profiles
comprise Rayleigh scattering from air molecules as well as scattering produced by ice particles when PMC are present. As
with ground-based lidars, we detect PMC by comparing the backscatter profile to a MSIS-E-90 total mass density profile
$\rho_{\text{ref}}(z)$ (Hedin, 1991) that is fitted to each lidar backscatter profile between 60 and 75 km altitude. Multiplication with the thus
determined normalization factor scales the backscatter profile to units of atmospheric density. The backscatter ratio $R(z)$ in
the altitude range betwen 76 and 90 km is obtained by division of the normalized backscatter profile by $\rho_{\text{ref}}(z)$. Finally, the
volume backscatter coefficient $\beta$ in units of $1/\text{m}/\text{sr}$ is determined by

$$\beta(z) = (R(z) - 1) \, \rho_{\text{ref}}(z) \, \frac{1}{M} \, N_A \, \sigma_R \tag{2}$$

with the molar mass of air $M = 28.8 \, \text{g} \, \text{mol}^{-1}$, the Avogadro number $N_A = 6.022 \times 10^{23} \, \text{mol}^{-1}$ and the Rayleigh scattering
cross section $\sigma_R = 6.32 \times 10^{-32} \, \text{m}^2 \, \text{sr}^{-1}$ for 532 nm wavelength (Thayer et al., 1995). In order to estimate the noise level, we
calculate the standard deviation $\sigma_{\text{bg}}$ of $\beta(z)$ between 88 and 90 km, a height where contamination from PMC ice particles is
unlikely. Values $\beta > 2.5 \, \sigma_{\text{bg}}$ are accepted as significant, and all others are masked.

Figure 3 illustrates the process of calculating density profiles from count data and subsequently deriving $\beta(z)$ for 11 July
2018, 1:35 UT for different resolutions, when a very weak PMC layer at 87 km resides above a bright and narrow double layer.
Although the background dominates the high-resolution profile down to about 70 km, the PMC signal in a thin layer at 83 km

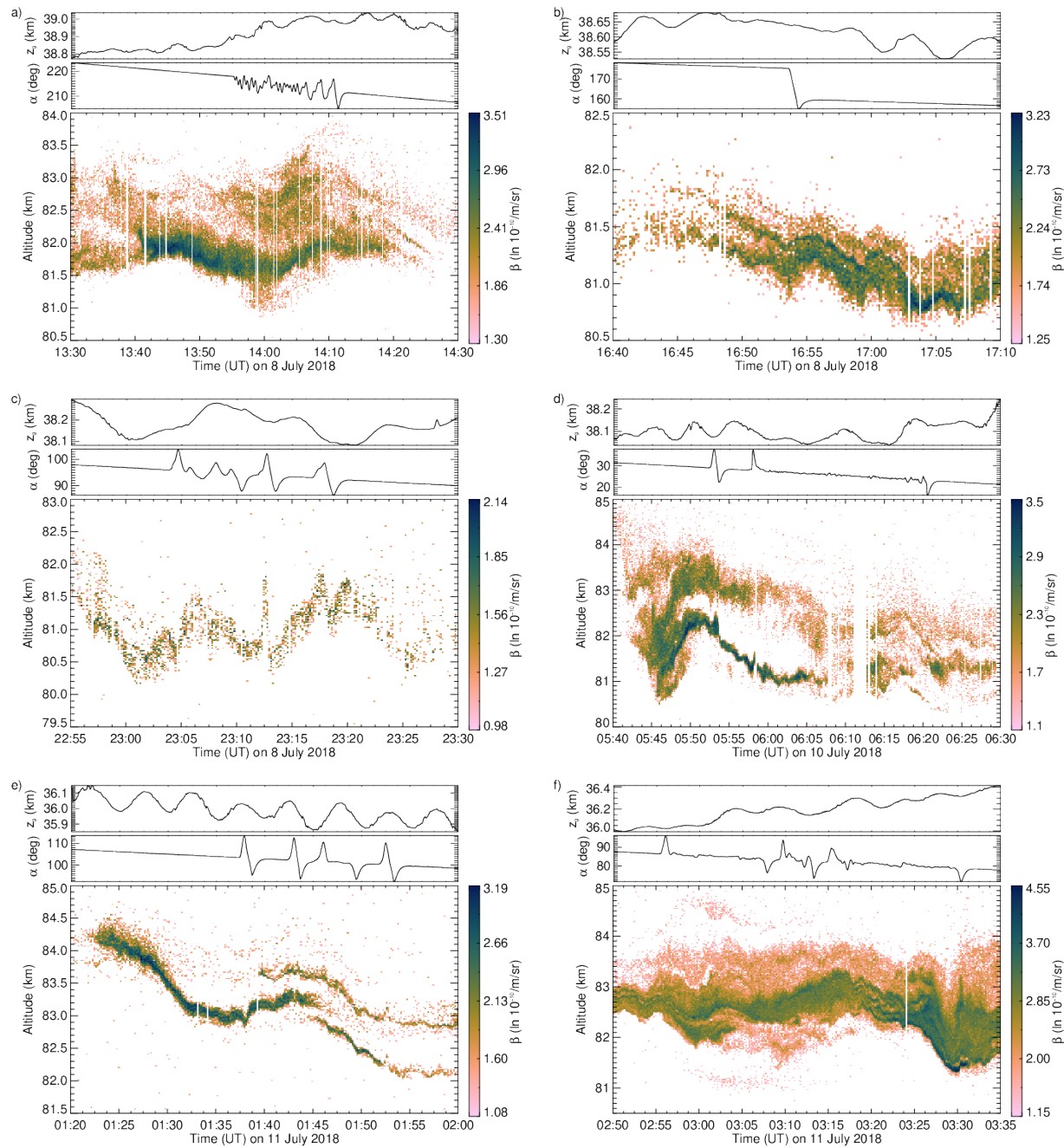

**Figure 2.** Floating altitude $z_g$, azimuth angle $\alpha$ and respective PMC detections for the six periods with deviations from the nominal rotator motion during PMC conditions. The floating altitude has been taken into account in PMC altitude calculations.

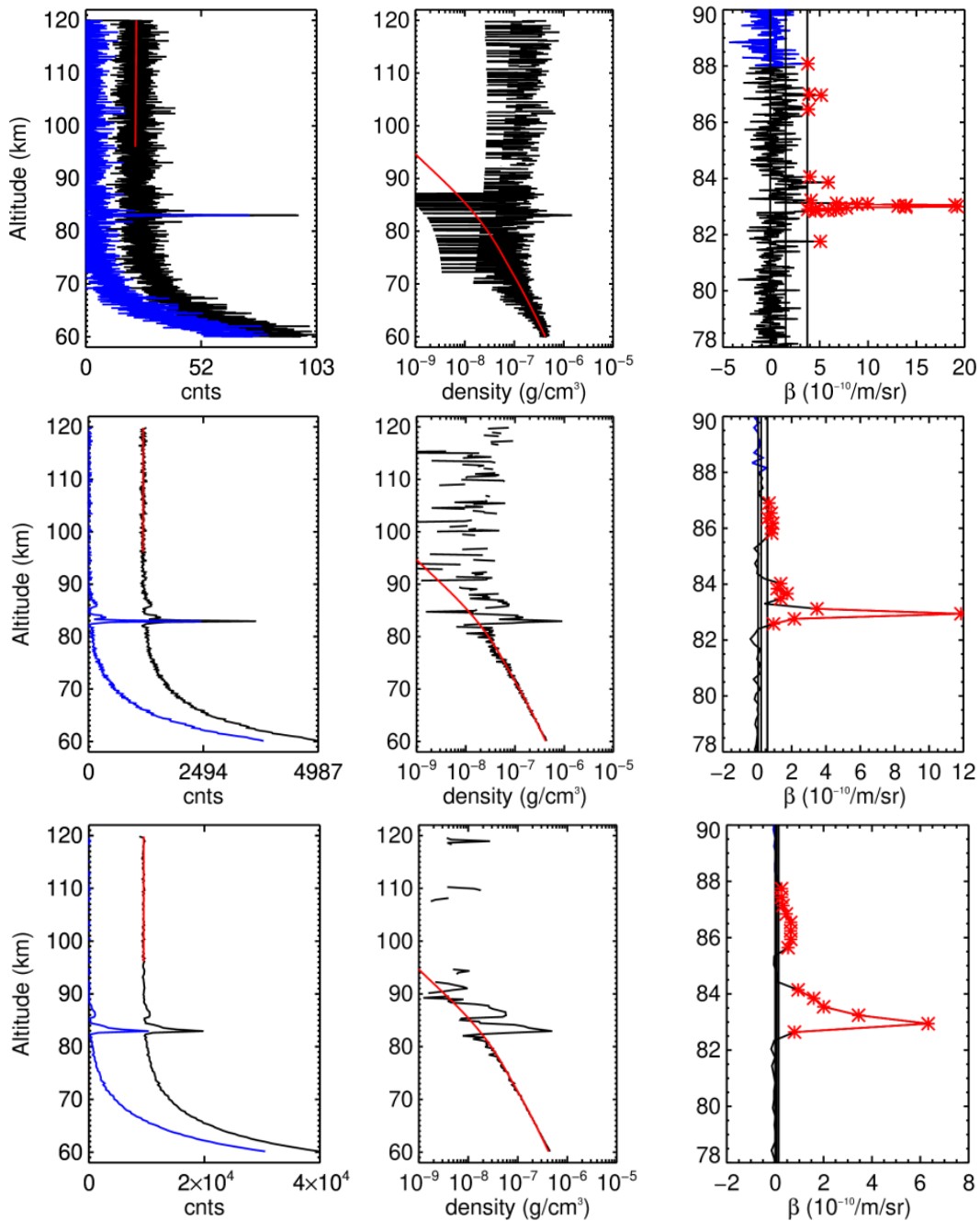

**Figure 3.** PMC signal centered at 11 July 2018, 1:35 UT, processed at three different resolutions 20 m × 10 s (top row), 180 m × 1 min (middle row), and 300 m × 5 min (bottom row). Left column: binned photon count profiles (black) and the same profiles with background removed (blue); the red line represents the background model fitted between 96 and 120 km altitude. Middle column: the calculated backscatter profile (black) and fitted MSIS-E-90 density profile (red). Right column: derived volume backscatter coefficients $\beta(z)$. The noise floor is estimated in the range 88–90 km (blue). Vertical lines indicate the mean noise, standard deviation $\sigma_{bg}$ and the used significance level 2.5 $\sigma_{bg}$. Values with $\beta(z) > 2.5\,\sigma_{bg}$ are marked in red.

is detected with high significance also at 10 s-resolution. The presence of a weak top layer is confirmed using lower-resolution data. While the detection threshold at the resolution of 20 m × 10 s is $2.5\,\sigma_{\rm bg} = 3.66 \times 10^{-10}/$m/sr, decreasing the resolution to 300 m × 5 min lowers the detection threshold to $2.5\,\sigma_{\rm bg} = 0.12 \times 10^{-10}/$m/sr. The signal-to-noise ratio increases as more signal is accumulated in larger bins. This comes at the cost of not resolving the fine details in PMC layer structure. As shown in the example, the maximum value $\beta_{\rm max} = 20 \times 10^{-10}/$m/s reduces to $\beta_{\rm max} = 12 \times 10^{-10}/$m/s and $\beta_{\rm max} = 7 \times 10^{-10}/$m/s

with decreasing resolution. This is due to the presence of sub-scale structure that is averaged out at lower resolution. The connection between resolution and signal-to-noise ratio also becomes evident when considering a five-hour period of PMC detections on 10/11 June 2018 that contains the profile selected in Fig. 3. Fig. 4a shows the faint, high-altitude layer at 87 km altitude that is detected at low resolution but not as clear at higher resolutions (Fig. 4b and c). On the other hand, the increase in resolution reveals smaller-scale structure that is only hinted at the lower resolution, e.g. the oscillation from 22 UT to

22:30 UT at 82 km altitude or the fine double-layer at 83 km at 1:45 UT. This revelation of smaller-scale structure yet only succeeds when the signal-to-noise ratio is sufficiently high, and thus $\beta$ is large enough. Also a lower background helps, but the background varies only by a factor of 2 during the day while $\beta$ varies over a much wider range.

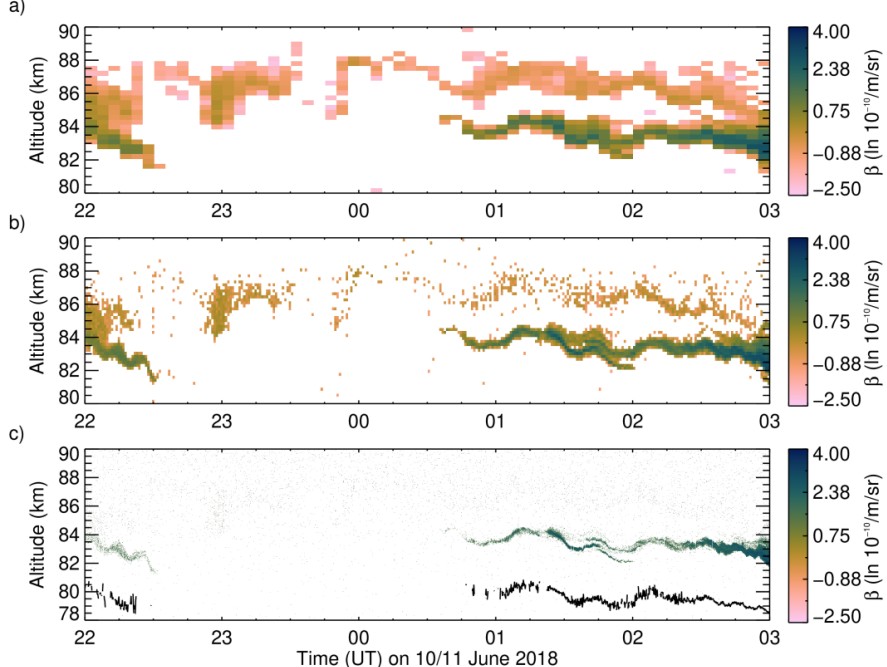

**Figure 4.** PMC soundings within the period 22–3 UT on 10/11 July 2018 for the three different vertical and temporal resolutions: (a) 300 m × 5 min, (b) 180 m × 1 min, and (c) 20 m × 10 s. The centroid altitude $z_c$ is added as a black line to (c), shifted down in altitude by 4 km.

To characterize the vertical displacements of the PMC layer, it is sometimes convenient to reduce the data to a one-dimensional time series, and a suitable representation of the PMC layer altitude is the $\beta$-weighted centroid altitude $z_c$. We



only evaluate $z_c$ where $\beta_{\mathrm{int}} = \sum_z \ln \beta(z) > 0.04 \times \ln 10^{-6}/\mathrm{sr}$. This effectively removes sporadic detections that cannot be attributed to a continuous PMC layer. To demonstrate this for an example that was specially selected for low $\beta$, we add $z_c$ as black curve to Fig 4c. The limit imposed on $\beta_{\mathrm{int}}$ does not significantly reduce the amount of usable data but ensures a high quality in the determination $z_c$. The resulting time series $z_c(t)$ can be subsequently interpolated and spectrally analyzed (the full time series of $z_c$ is shown in Fig. 5d). In order to avoid having to make further assumptions on the coherency of the PMC

layer, we refrain from applying additional spatial filters.

## 4   Statistics

PMC Turbo was at floating altitude for 138.61 h (5.77 d) between 10:10 UT on 8 July and 4:46 UT on 14 July 2018. The exclusion of the periods during which the lidar was not collecting data due to tests of the instrument or software problems leaves 132.4 h of nominal operation time with high-quality data, accounting for more than 95 % of the flight time. The flight

of PMC Turbo took place 17–22 days from solstice, when the occurrence of PMC is generally maximum. PMC are detected during 49.7 h. The light grey boxes overlaying the time series of $\beta_{\mathrm{int}}$ in Fig. 5b mark periods with PMC detections. The resulting occurrence rate is 36.8 %. Fritts et al. (2019) show statistics derived from the Cloud Imaging and Particle Size (CIPS) instrument onboard the NASA Aeronomy of Ice in the Mesosphere (AIM) satellite proving that the observed PMC occurrence rate was average with respect to the time of season, latitude and previous years.

The time- and range-resolved PMC detections are shown in Fig. 5d. From a Gaussian fit to the flight-mean $\beta$ profile (Fig. 5f), we derive a mean PMC altitude of 82.43 km with a standard deviation of 1.08 km. This is in agreement with satellite data for this latitude band of 82.6 km reported by Carbary et al. (2001). The measured mean altitude is 840 m lower but within uncertainties intervals compared to $83.27 \pm 1.30$ km reported by Fiedler et al. (2017) based on the multi-year ALOMAR Rayleigh/Mie/Raman lidar dataset at 69°N, 15°W. Unlike the ALOMAR dataset, the BOLIDE dataset was obtained at a

different range of longitudes (21°W–110°E compared to 15°W) and is of short duration, and thus likely susceptible to sampling biases induced by tidal, seasonal and intra-annual variation.

The $\beta$ values show an exponential distribution covering a wide range from $10 \times 10^{-10}/\mathrm{m/sr}$ to $80 \times 10^{-10}/\mathrm{m/sr}$. The histogram shown in Fig. 6 includes all values of $\beta$ between 79 and 86 km altitude. A linear fit is applied to the data and shown shifted to larger values for better visibility in Fig. 6 by a black line. Its slope $\alpha_{\mathrm{BOLIDE}} = -0.1086 \pm 0.0005$ is in agreement

with results from the ALOMAR dataset of $\alpha_{\mathrm{ALOMAR}} = -0.1079 \pm 0.0045$ between $3$–$50 \times 10^{-10}/\mathrm{m/sr}$ based on $\beta_{\mathrm{max}}$ at 15 min temporal resolution (Berger et al., 2019).

An overview of the variability of the PMC layers in the BOLIDE dataset is shown in Figs. 7 and 8. Periodic motions of the PMC layer altitude with periods of few hours down to the buoyancy frequency of $\approx 5$ min are induced by gravity waves. The BOLIDE dataset also shows other, non-linear large-scale phenomena, e.g. mesospheric bores, of which the two occurrences

on 13 July at 12:30 and 13:30 UT (Fig. 8i) were analyzed in detail by Fritts et al. (2020). In the spatial and temporal vicinity of such structures as well as when gravity waves break, a variety of instability dynamics is induced. Here, the true advantage of the BOLIDE dataset comes to play, as it is only at high temporal and vertical resolution that these dynamical processes are

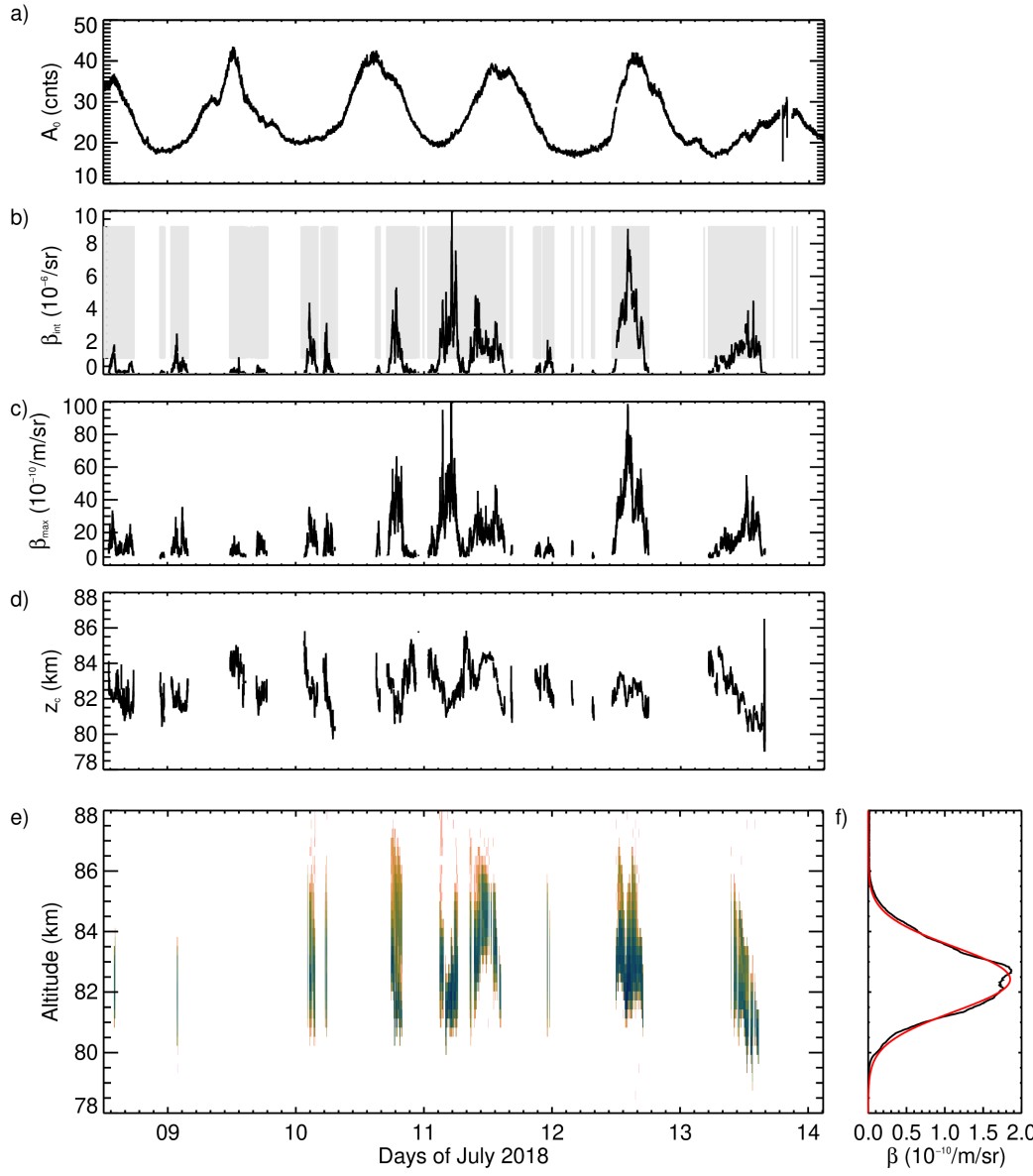

**Figure 5.** (a) $A_0$ representing the background at 96–120 km that is removed from the photon count profiles, (b) vertically integrated volume backscatter coefficient $\beta_{\mathrm{int}}$, (c) maximum volume backscatter coefficient $\beta_{\max}$, (d) centroid altitude $z_c$, (e) time-altitude section of volume backscatter coefficients $\beta$ and (f) flight-mean $\beta$ (black line) as a function of altitude, and a Gaussian fit (red line) with a mean of 82.43 km and standard deviation of 1.08 km. The resolution of the data is 20 m and 10 s.

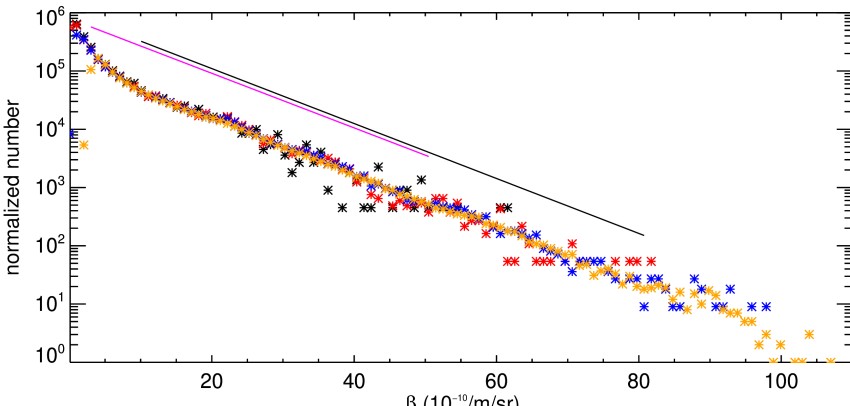

**Figure 6.** Histogram of volume backscatter coefficients $\beta$ between 79 and 86 km altitude at 300 m $\times$ 5 min (black), 180 m $\times$1 min (red), 60 m $\times$ 30 s (blue) and 20 m $\times$ 10 s (orange) resolution, scaled to account for the different number of bins. The slope $\alpha_{\mathrm{BOLIDE}} = -0.1086 \pm 0.0005$ of the black line was fitted to the data in the range $\beta = 10$ to $80 \times 10^{-10}$/m/sr. The magenta line with $\alpha_{\mathrm{ALOMAR}} = -0.1079 \pm 0.0045$ was determined from the ALOMAR RMR lidar dataset by Berger et al. (2019).

resolved. At short time scales of 2 min and less, locally enhanced $\beta$ results mainly from convergence of air masses that increase the local ice number density and not so much from ice particle growth, as the latter is a slower process. However, ice particles

may quickly sublimate during rapid downdrafts. To interpret specific dynamics observed by the lidar, the common-volume PMC Turbo images are very helpful as they provide the spatial context (viewed in two dimensions from below) in the direct neighborhood of few hundred meters as well as information on the structure and evolution of the larger-scale cloud field up to $\approx 160$ km distance (Kjellstrand et al., 2020, , their Fig. 5). For example, the largest value of $\beta$ exceeding $100 \times 10^{-10}$/m/sr occurred after a rapid decrease of the PMC lower boundary on 11 July at 3:30 UT (Fig. 8f) and is related to the passage of a

large-scale vortex ring of several km diameter. This and other types of patterns at the scale of few minutes and less that are included in the BOLIDE dataset and typically occur in lidar soundings of PMC layers are studied in more detail by Kaifler et al. (submitted in May 2022). In-depth analysis including modelling of events during PMC Turbo concern the small-scale vortex rings on 10 July at 2:40 UT (Fig. 7d, Geach et al., 2020) and the Kelvin-Helmholtz instability dynamics on 12 July at 13:40 UT (Fig. 8h, Fritts et al., in review May 2022; Kjellstrand et al., in review May 2022). An analysis of the dynamics of

breaking gravity waves during the bright PMC displays on 11 July between 4 UT and 6 UT (Fig. 8f) is in preparation.

## 5  Conclusions

The BOLIDE dataset available from zenodo and NASA's Space Physics Data Facility contains PMC volume backscatter coefficients at 20 m $\times$ 10 s resolution. Additionally, time series of balloon altitude, azimuth angle and beam position at 82 km altitude are provided. The dataset as described in this document is suitable for analysis of small-scale dynamical processes

acting on the PMC layer, especially in combination with the images acquired by the PMC Turbo wide- and narrow-field-of-



**Figure 7.** Part I of selected PMC altitude-time sections which are part of of the PMC Turbo BOLIDE dataset. Short gaps in the lidar soundings are due to missing data.

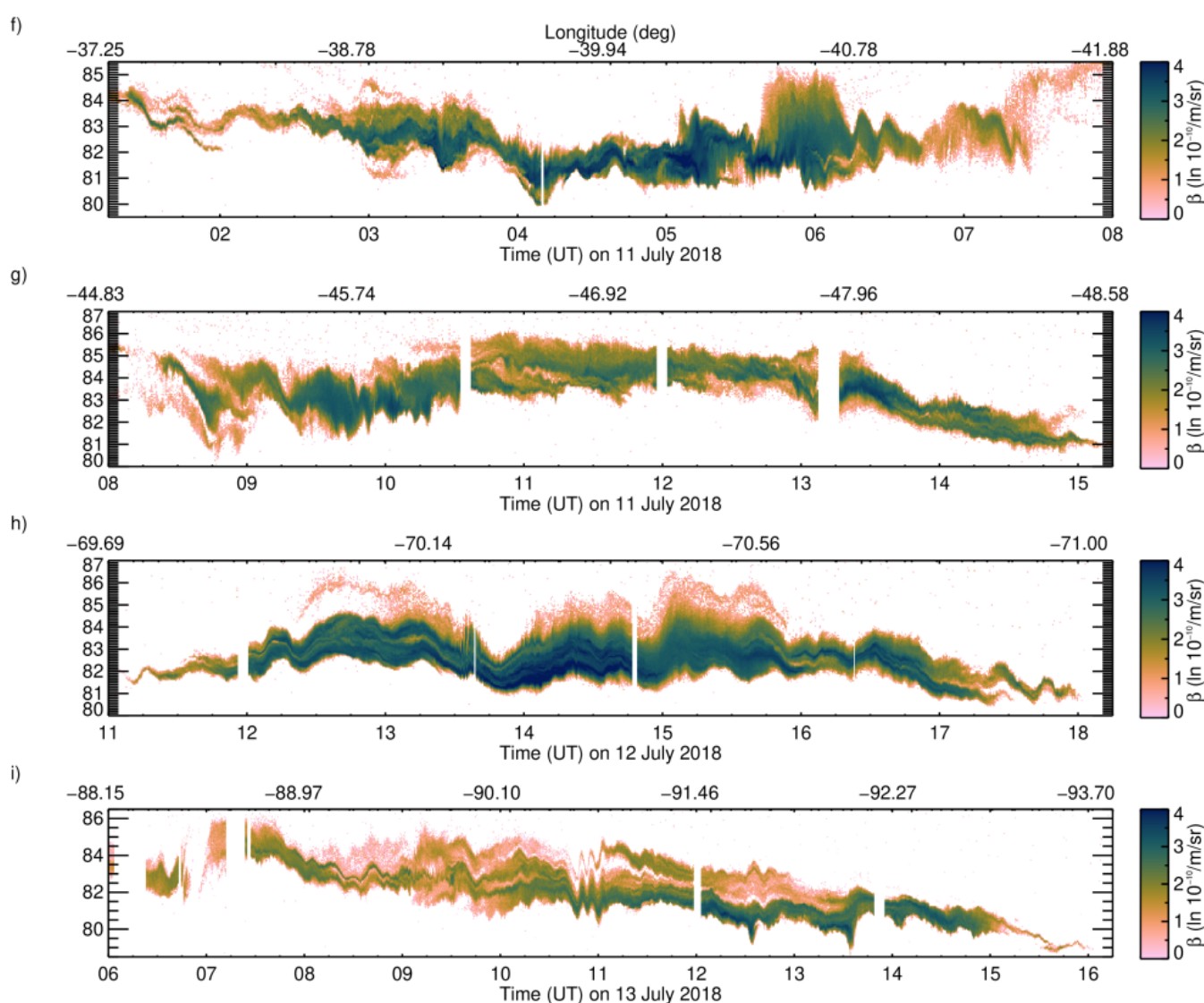

**Figure 8.** Part II of PMC altitude-time sections which are part of the PMC Turbo BOLIDE dataset.



view cameras. The limited flight duration of 5.9 days likely results in sampling biases due to tidal and seasonal variation, affecting e.g. the flight-mean PMC altitude. We thus hope to obtain a larger data set from the upcoming Balloon Sodium Lidar to measure Tides in the Antarctic Region (B-SoLiTARe) mission which is currently scheduled for launch at McMurdo, Antarctica, in 2024 and will provide a flight time of approximately three weeks. The analysis of the data described here can be

applied to future datasets acquired with air-borne (balloon or aircraft) or ground-based Rayleigh lidar instruments.

## 6   Data availability

The dataset described here containing BOLIDE volume backscatter coefficients $\beta$ at 20 m $\times$ 10 s resolution are available as netcdf file from https://zenodo.org/record/5722385 (Kaifler, 2021). A copy is hosted by NASA's Space Physics Data Facitily at https://cdaweb.gsfc.nasa.gov/ under section "Balloons".

*Author contributions.*   NK wrote the manuscript, analyzed the data, and prepared the figures and the netcdf files. BK designed and built the instrument and operated it during the flight. MR obtained funding for the instrument. DF is the mission PI for PMC Turbo.

*Competing interests.*   The authors declare no competing interests.

*Acknowledgements.*   We acknowledge the efforts of the PMC Turbo team in building and integrating the PMC Turbo gondola and payload, successfully operating it during flight and the cooperation during data analysis. Robert Reichert assisted with the remote control of the lidar

instrument. CSBF is acknowledged for flight operations and logs of GPS data. Colour maps used in this work are from Crameri et al. (2020). We thank the Space Physics Data Facility for hosting the dataset. It has been a privilege to take part in this NASA mission.





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
