# Peer review of "The polar mesospheric cloud dataset of the Balloon Lidar Experiment BOLIDE"

_Earth System Science Data, 2022_

## Referee Comment (RC1)

**Manuscript by Kaifler et. al:**
**The polar mesosphere cloud dataset of the Balloon Lidar Experiment BOLIDE**

General comments:

The manuscript describes PMC data obtained from a stratospheric balloon flight during a 6 day mission from Sweden to Canada. PMC are known since more than 100 years as noctilucent clouds (NLC) and consist of nanometer sized ice particles in an altitude range from roughly 80 to 90 km. They occur during summer at polar latitudes due to low temperatures and enhanced water vapor at this altitude region. Besides their role in the field of climate change, PMC are important tracers for the dynamics at such high altitudes. The ice particles visualize changes in ambient temperature and water vapor as well as show imprints of atmospheric waves at different time scales. In recent years, high resolution lidar measurements (spatial and temporal) of PMC contribute increasingly to the understanding of small-scale dynamical processes induced by the breaking of atmospheric gravity waves and their energy transition into turbulence.

BOLIDE contributes to this topic as balloon-based lidar by profiling PMC layers over an extended longitude section, compared to fixed location ground-based lidars. Results of this experiment have been already published, indicating the significance of the dataset. Here, the authors describe the primary parameter of the lidar soundings (time and range resolved volume backscatter coefficient), along with position data of the ballon. Additionally some mean properties of the dataset are compared to multi-year ground-based lidar data at ALOMAR to confirm general consistency. Special emphasis is put on the determination of altitude and orientation of the moving lidar, which is essential for the interpretation of the observations. In my opinion the manuscript addresses all necessary topics to assist potential users of this dataset in understanding the processing chain.

The dataset is available for download at Zenodo. I was able to extract the data from the NetCDF file. Some quick plots confirmed information given in the manuscript.

The manuscript is well written and structured. The data are of importance for the investigation of high altitude dynamical processes, which are hardly accessible by other experimental methods.
I recommend publication in ESSD after addressing the following specific comments.

Specific comments:

Line 35:
  • The tilde sign should not be separated from the number.

Section 2:
  • I could imagine that there was a gondola swing and wonder about its impact on the laser beam direction, and hence the spot inside the PMC layer (in addition to the rotation). Please clarify this.

Figure 1:
  • Please increase the width of the plots, the significance values in the lower panels are hardly readable.

Lines 95-97, Fig. 2:
- It is good practice to make data users aware of potentially strange data periods. However, showing the belonging plots as first PMC plots in the course of the manuscript appears unwise. I suggest putting Fig. 2 into an appendix and adding some short text to make aware, e.g., that double layers might be caused by 2 single layers at different locations (or something like that).

Line 118:
- Why is A0 influenced by PMC? The background is determined above 96 km, i.e. well above the PMC layers. Please explain.

Line 144:
- … only hinted at at the … (typo)

Figure 4:
- It would be better to have identical y-scalings for all plots.

Line 150:
- Please check if logarithm in the formula for BETAint is correct.

Line 169 and thereafter:
- The longitude of ALOMAR is 16 deg E.

Line 171:
- "intra-annual"; Do you mean year-to-year variations? If yes, please use "interannual".

Figure 5:
- I miss a first part of the caption, describing what is basically shown here.
- Please explain the gray areas (panel b) in the caption.
- I wonder why there are no BETA values at times BETAmax exists (panel e vs panel c). Please check.
- Please add a color bar for panel e.
- I wonder about the low mean BETA values in panel f, even though the data resolution is highest. Fig. 4 shows that at this resolution low BETA values are not significant and thus missed. Hence I would expect larger mean values. Please explain.

Figures 7 and 8:
- Please add vertical axis labels and reduce minor tick numbers in some panels.

Line 260:
- The title is: "Auffallende Abenderscheinungen am Himmel".

Line 262-263:
- The paper is already published.

---

## Author Response (AR1)

**Dear Editor,**

Below we include our response to all the reviewer's comments that include descriptions of the changes made to the manuscript. The manuscript submitted to ACPD is now available as a preprint with DOI and the citation was updated. In addition, we have changed the color map from batlow to viridis.

Natalie Kaifler

**Reviewer #1**

We thank the reviewer for acknowledging the potential of this type of data for atmosphere dynamics studies and the positive review.

On the impact of a gondola swing (section 2): It is true that the buoyancy motion of the balloon also induces a swing of the gondola that affects the zenith angle of the laser beam pointing. We investigated this effect by determining the angle of the horizon in images taken with a side-viewing camera. More accurately, the swinging can be quantified by evaluating the star positions in the PMC Turbo camera images. A validated astrometry data product is still in work, but based on the horizon measurements we estimated the uncertainty induced on the laser beam zenith angle to be less than 0.1 degree on average. This is the uncertainty given in l. 77, translating to 80 m horizontal distance within the PMC layer (l. 82). We slightly modified the sentence in line 77 to: "... was pointed 28 deg off-zenith, with an uncertainty of about 0.1 deg caused by the mentioned buoyancy oscillations."

On the influence of PMC on the background (l. 118): It is true that the background is determined well above the PMC layers. There is however an indirect effect of PMC on the background, in the sense that the sky brightness increases when PMC are present. The background is thus not solely determined by the solar zenith angle (that is determined by local time), as is evident in Fig. 5a. For high altitudes where there is no contribution from scattered laser light any more, the lidar works like a photometer that measures the sky brightness at the position of the gondola. The following text was added to the revised manuscript: "... but is also influenced by PMC as they increase the sky brightness, too."

In line 150 we described our analysis in words in order to prevent confusion about the units, and to clarify how this was implemented. The text was changed to: "We only evaluate $z_c$ for profiles where the sum of all ln beta values exceeds a value of 0.04."

We added "Evolution of PMC parameters during the 6-day flight of PMC Turbo:" to the caption of Fig. 5. The grey areas in Fig. 5b were supposed to aid a visual estimation of PMC occurrence frequency: grey areas are with PMC detections and white areas are without. This was mentioned in the text in line 162, but was missing from the caption, so we added ", with the light grey areas indicating times with finite beta_int for a better visual impression of PMC occurrence frequency, ".

The problem with Fig. 5e not showing beta values at times where beta_max exists was possibly an image resolution problem. The panel was replaced with a high-resolution plot of the data, and panel e and c now agree with regard to detection times as shown below.

The beta profile in Fig. 5f was a flight-mean profile normalized by the total flight time and thus includes periods of time with no PMC detections. Therefore, the absolute values were lower. We agree that it is more common to normalize by periods with PMC detections, and the updated curve exhibits a maximum of beta=5, as shown below.

[Figure]

We thank the reviewer for spotting errors that we overlooked: additional blanks (l. 35), wrong longitudes (l. 169), wrong titles in the citations (l. 260) and citations that have been updated in the meantime (l. 262-263). This has all been corrected in the revised manuscript. L. 171 was changed to "interannual". L. 144 "to hint at sth." is, we believe, not a typo, but a copy editor may help.

In Fig. 1, the width was increased as suggested, and the overlapping labels for 95% significance removed; instead, the information is included in the caption, which makes the figure better to read without losing information. Fig. 2 was put into an appendix as suggested, improving the readability. Fig. 4 was changed to identical y-scalings as suggested. A color bar in Fig. 5e was added. In Fig. 7 and 8 the vertical axis labels have been added and the minor tick numbers reduced as suggested.

**Reviewer #2:**

Two of the manuscripts which we cited and were not yet available online at the time of review are now published. The two companion papers on multi-scale Kelvin-Helmholtz instability dynamics observed by PMC Turbo on 12 July 2018 can be found at

https://agupubs.onlinelibrary.wiley.com/doi/abs/10.1029/2021JD036232

https://agupubs.onlinelibrary.wiley.com/doi/abs/10.1029/2021JD035834

and the updated citations are:

Kjellstrand, C. B., Fritts, D. C., Miller, A. D., Williams, B. P., Kaifler, N., Geach, C., Hanany, S., Kaifler, B., Jones, G., Limon, M., Reimuller, J., and Wang, L.: Multi-Scale Kelvin-Helmholtz Instability Dynamics Observed by PMC Turbo on 12 July 2018: 1. Secondary Instabilities and Billow Interactions, Journal of Geophysical Research: Atmospheres, 127, e2021JD036 232, https://doi.org/https://doi.org/10.1029/2021JD036232, e2021JD036232 2021JD036232, 2022.

Fritts, D. C., Wang, L., Lund, T. S., Thorpe, S. A., Kjellstrand, C. B., Kaifler, B., and Kaifler, N.: Multi-Scale Kelvin-Helmholtz Instability Dynamics Observed by PMC Turbo on 12 July 2018: 2. DNS Modeling of KHI Dynamics and PMC Responses, Journal of Geophysical Research: Atmospheres, 127, e2021JD035 834, https://doi.org/https://doi.org/10.1029/2021JD035834, e2021JD035834 2021JD035834, 2022.

The manuscript by Kaifler et al. on "Signatures of gravity wave-induced instabilities in balloon lidar soundings of polar mesospheric clouds" submitted to ACPD under acp-2022-572 is currently awaiting reviewer's reports and are hopefully available online within few days.

*"If possible, a brief description of data analysis methods on how to use your dataset to obtain small-scale features (such as vortex rings, instability structures, and so on) would be helpful for readers. "*

The data can be subjected to a variety of analysis methods that are suitable to detect patterns like correlations, feature detections or spectral methods. The chosen method will likely depend on the specific goal of a study and may also depend on the case, i.e. the characteristics of the event to be studied. For guidance, the cited publications analyzing PMC Turbo or ALOMAR RMR lidar data are useful. It would be desirable to be able to define the identified small-scale features well enough such that they can automatically be pulled out of a lidar dataset by a standard algorithm. Possibly, the natural variability and different viewing geometries and wind speeds make it necessary to evaluated the dynamics case by case. The work by Kaifler et al. submitted to ACPD mentioned above employs a general method based on  the evaluation of gradients at high resolution to locate small-scale features, and then discusses examples that are grouped by morphology and the likely underlying dynamics, for which tailored methods can be developed in the future.